# Hydraulic-driven adaptable morphing active-cooling elastomer with bioinspired bicontinuous phases

Dehai Yu[1,2], Zhonghao Wang [1,2], Guidong Chi[1], Qiubo Zhang[1], Junxian Fu[1], Maolin Li[1], Chuanke Liu[1], Quan Zhou[1], Zhen Li[1], Du Chen[1], Zhenghe Song[1] & Zhizhu He [1] ✉

The active-cooling elastomer concept, originating from vascular thermo-regulation for soft biological tissue, is expected to develop an effective heat dissipation method for human skin, flexible electronics, and soft robots due to the desired interface mechanical compliance. However, its low thermal conduction and poor adaptation limit its cooling effects. Inspired by the bone structure, this work reports a simple yet versatile method of fabricating arbitrary-geometry liquid metal skeleton-based elastomer with bicontinuous Gyroid-shaped phases, exhibiting high thermal conductivity (up to 27.1 W/mK) and stretchability (strain limit >600%). Enlightened by the vasodilation principle for blood flow regulation, we also establish a hydraulic-driven conformal morphing strategy for better thermoregulation by modulating the hydraulic pressure of channels to adapt the complicated shape with large surface roughness (even a concave body). The liquid metal active-cooling elastomer, integrated with the flexible thermoelectric device, is demonstrated with various applications in the soft gripper, thermal-energy harvesting, and head thermoregulation.

Just as the body-core temperature of healthy people should be maintained around 37 °C, the appropriate temperature is one of the critical factors in achieving reliable and lasting operation of electronic devices, especially since overheating of wearable electronics even induces skin injury. Accordingly, developing effective heat dissipation methods capable of conformal morphing, high cooling effects, and long-term operation has played a crucial role in personalized thermal management[1,2], clinical practice[3,4], soft robotics[5–7], flexible electronics[8–10], and thermal energy harvesting[11]. Newton's law of cooling, presented by $q = C_{ht}\Delta T$ ($\Delta T = T_b - T_a$ denoting the difference in the temperature of the body and the ambient), has indicated that the body temperature is mainly determined by the heat transfer coefficient of $C_{ht}$ for the given heat generation of $q$. However, the passive cooling methods (including the natural convection and radiative effects) have a low $C_{ht}$ (<10 W/m²K) for the human skin in the indoor environment[12].

Although the newly emerging passive cooling method (such as radiation[13], thermal absorption through the phase change of liquid-solid[14] or liquid-vapor[15] transition) is energy efficient (no required energy input) and lightweight, its limited cooling/heating capacity makes it hard to meet the long-term stable operation, especially for the high-power electronics device[16], extreme ambient temperature[17] or clinical requirements[18]. The active cooling method based on the fluidic-channel heat sink has a large $C_{ht}$ (>100 W/m²K)[19], which even achieves a lower temperature than the surrounding ambient temperature by combining it with the thermoelectric device[20]. However, its rigidity-induced poor adaptation limits heat dissipation capacity for application to the soft body.

Inspired by the vascular system carrying blood throughout the soft tissue to remove the metabolic heat generation and maintain body-core temperature, the flexible active-cooling elastomer (ACE)

[1]Center for Agricultural Flexible Electronics Technology, College of Engineering, China Agricultural University, Beijing 100083, China. [2]These authors contributed equally: Dehai Yu, Zhonghao Wang. ✉e-mail: zzhe@cau.edu.cn

solution by embedding the thermo-fluidic channel into the silicone elastomer matrix enables excellent deformation performance due to its high stretchable capacity (strain limit larger than 800%). It is noteworthy that the large-tensile performance of ACE does not guarantee its adaptation to curved surfaces (such as the concave part), especially for dynamically changing shapes. Passive deformation cannot address this critical issue, limiting its wide application in personalized thermal management and flexible electronics. Although the silicone elastomer has good resistance to chemical agents, heat, weathering, and ultraviolet irradiation, its low intrinsic thermal conductivity (~0.2 W/mK) severely blocks its heat conduction[21,22]. Room-temperature gallium-based liquid metal (LM) embedded elastomer (LMEE)[23] has been recently demonstrated to improve heat conduction through stretching while less impacting its stretchable capacity due to the LM liquid feature[24-29]. However, the thermal conductivity of LMEE is only 1.6 W/mK for a large LM volume fraction of $\Phi_{LM} = 50\%$ under stress-free conditions, which even has a slight decrease (<1.6 W/mK) in the orthogonal directions when strained to 400%. Increasing the LM volume fraction cannot effectively improve its heat conduction but weakens its stretchability, such as a low strain limit of 75% for LMEE with $\Phi_{LM} = 80\%$[30]. Another challenge is fabricating complex thermo-fluidic channel structures in silicone elastomers, simultaneously achieving high heat conduction and stretchability. The uncured liquid silicone with low viscosity could be printed to a 3D structure through particle filler-modified viscosity or well-designed support material[31]. However, most 3D-printed silicone-elastomer geometry embedded with channels presents low thermal conductivity, which cannot achieve efficient cooling effects. The complex-shaped structure (such as embedded thermo-fluidic channels) based on LMEE[30,32] is hard to obtain through 3D printing, which also has a low thermal conductivity. The sugar-based cast molding method has also been used to form percolating LM networks in elastomer[33,34], but it cannot achieve complicated shapes.

Inspired by the bicontinuous structure of bone tissue, this work reports a simple yet versatile method of fabricating arbitrary-geometry LM skeleton (LMS)-based ACE (LMS-ACE) using fused deposition modeling (FDM)-printed acrylonitrile butadiene styrene (ABS) dissolvable mold with the designable infill path/rate (Supplementary Movie 1), which successfully obtains complex embedded channel architectures, especially for the structure with inner completely isolated hollow cavities. The bicontinuous Gyroid-shaped microstructures are designed to enable LMS-ACE to achieve a high thermal conductivity (21.2 W/mK at zero strain and 27.1 W/mK at 300% strain) and stretching (strain limit >600% and stiffness <100 kPa), exceeding other soft elastomer-based ACE, even better than stainless steel-based rigid heat sink (~16 W/mK). Enlightened by the vasodilation principle for blood flow regulation in biological systems, we also build a hydraulic-driven conformal morphing strategy for better thermoregulation by modulating the hydraulic pressure of channels to adapt the complicated shape with large surface roughness (even a concave body), which can reduce contact thermal resistance and increase thermo-fluidic flow flux in the channel, and thus enhance the cooling effects. The LMS-ACE combined with the flexible thermoelectric device (FTED) is then demonstrated with various applications, including the soft gripper with hydraulic-driven conjoint functions of actuation and active cooling, thermal-energy harvesting from the curved surface and thermoelectric conversion enhanced by hydraulic-driven adaptable morphing, and wearable flexible headband with high cooling capacity (temperature drop >20 °C) and long-term operation (>24 h).

## Results
### Fabrication of 3D complex-shaped LMS-ACE
The poor adaptation of rigid heat dissipation (Fig. 1A-I) limits its application in flexible electronics, personalized thermal management,

and soft robotics. Although the ACE presents good flexibility, its low heat conduction and passive adaptation (Fig. 1A-II) cannot achieve gratifying cooling effects. Inspired by the bicontinuous structure of bone tissue, our proposed LMS-ACE (Fig. 1A-III) with a synergistic combination of metal-like thermal conductivity and high stretchability could address the above critical issues through an active thermo-fluidic hydraulic-driven adaptation strategy. However, it is challenging to fabricate the continuous open-porous LMS with a complicated 3D free-standing shape due to its high surface tension and liquid state at room temperature. The traditional cast molding method (such as the sugar-particle sacrificial template) is only suitable for preparing the simple LMS structure with limited porosity (20%–50%), especially hard to achieve complicated shapes with inner hollow cavities (such as thermo-fluidic channels). Here, we present a design method of 3D printing the porous ABS-based geometry as the desired sacrificial template using a commercially available desktop FDM printer, followed by perfusing with the LM through vacuum infiltration, freezing to solid state at low temperature (such as −20 °C), and removing the template through the dissolution of ABS in dichloromethane (DCM) (Fig. 1B). Compared with other fabrication methods of direct writing, inkjet printing and stereolithography[35], the ABS-based FDM not only prints the complicated geometry and also allows the model to be easily dissolved through non-thermal dissolvable effect. The critical design principle for obtaining a complex-shaped porous negative model is to coordinately print a continuous open-porous structure and the dense support part, which of the former must allow the LM perfusion inside and quite the contrary for the latter. In addition, the exposed surface of the porous part at least should be enclosed by a thin wall (about two layers), which could prevent the LM from perfusing out. The layer-by-layer printing could easily control spacing width ($W_s$) between adjacent traces and obtain the designed porosity (corresponding to the LM perfusion volume ratio) with an approximate estimation of $\Phi_{LM} = (W_s + 0.21 h)/(W_s + W_l)$, where $W_l$ denotes the extruded fiber line width (often smaller than the extrusion nozzle diameter of 200 μm due to drawing shrinkage effect) and $h$ for its height (Supplementary Fig. 1). Notably, $\Phi > 0$ for $W_s = 0$ is attributed to the gap between the stacked cylindrical fiber, which produces the loose support structure to allow the LM perfusion inside. Our test demonstrated that the optimal printing parameters of the low layer height ($h = 20$ μm), the high extrusion temperature (>280 °C), the overlap ($W_s \approx -0.21 h$) of the adjacent traces, and the low printing speed (<40 mm/s) could enhance the interlayer bonding strength (Supplementary Fig. 2), leading to a tight structure and thoroughly preventing LM from perfusing into the support part (Supplementary Fig. 3). Such a design successfully obtains the complicated LMS geometry (Fig. 1C), especially for the structure with inner completely isolated hollow cavities (Supplementary Fig. 4). The LMS elastomer with the bicontinuous structure is then easily prepared by dip-coating silicone due to the capillary effect of porous structures, where the coated thickness is controlled by the cycle-dipping times (Supplementary Fig. 5). The proposed method could even prepare the LMS with graded porosity and lightweight geometry with an effective density of 0.45 g/cm³, corresponding to $\Phi_{LM} = 7.7\%$ (Supplementary Fig. 6). The superhigh stretching and thermal conduction of the LMS-ACE enable fluidic elastomer actuator with excellent cooling performance through integration with the FTED (Methods and Supplementary Fig. 7). The cooling capacity of FTED strongly depends on the heat removal at its hot side. Thus, the FTED used as the inextensible layer of the gripper is bent and releases its hot-side heat when the water is pressurized to flow through the chambers of the extensible layer (Fig. 1D and Supplementary Movie 2). Our soft gripper with a cooling function could reduce the object's surface temperature from 79.7 °C to −2.4 °C within 15 min (Fig. 1E), which is significantly lower than the ambient temperature (22 °C).

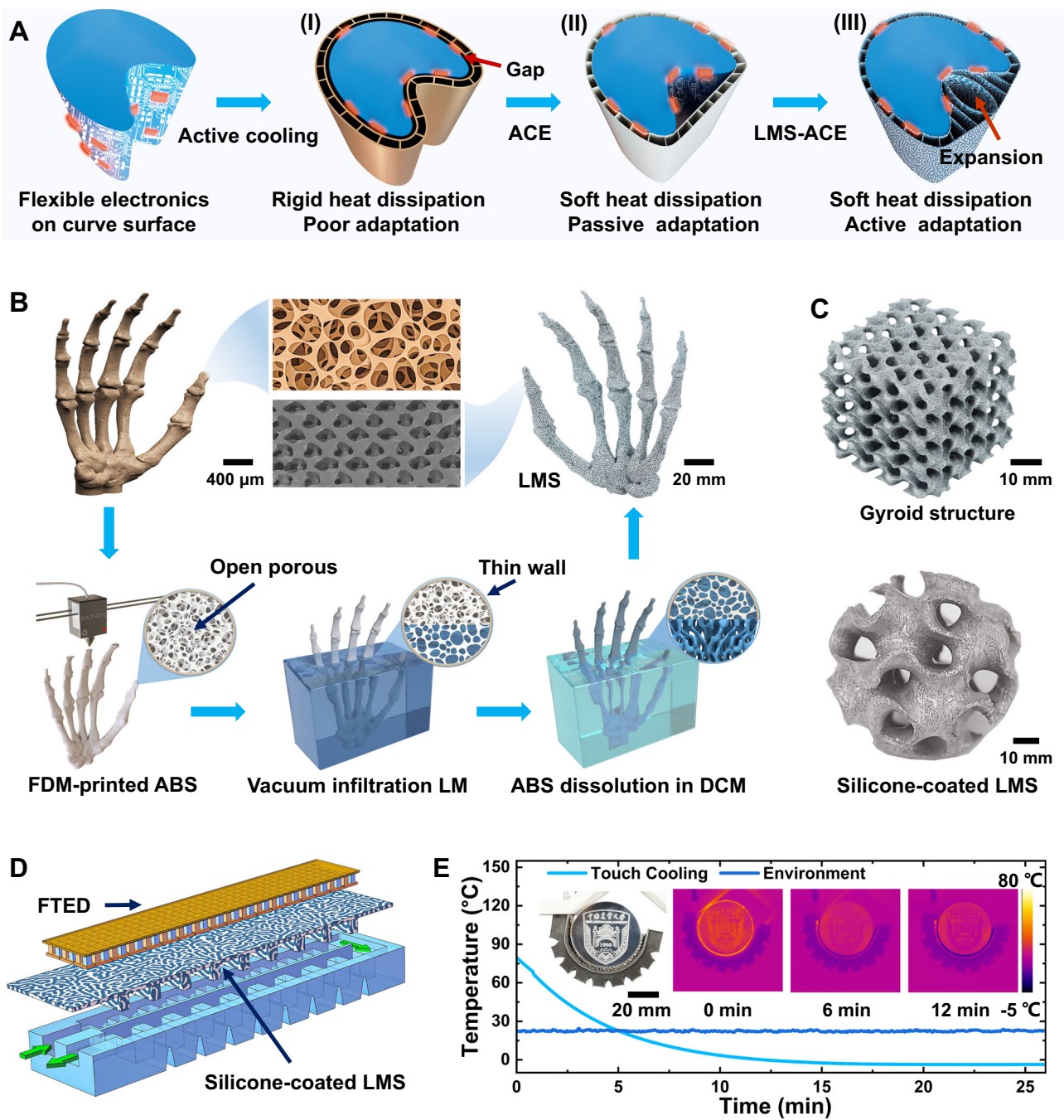

**Fig. 1 | Fabrication of 3D complex-shaped liquid metal skeleton-based active-cooling elastomer (LMS-ACE). A** Illustrations of the different active cooling methods for flexible electronics on the curved surface, allowing dynamically changing shapes: (I) rigid heat dissipation device with high thermal conduction but poor adaptation, (II) ACE with low thermal conduction and passive adaptation, (III) LMS-ACE with high thermal conduction and active adaptation. **B** Fabrication principle of the LMS with internetwork structure using fused deposition modeling (FDM)-printed acrylonitrile butadiene styrene (ABS) dissolvable mold: 3D printing the porous ABS geometry, perfusing with the LM through vacuum infiltration and dissoluting ABS in dichloromethane. **C** Complex-shaped LMS and its coating with silicone elastomer. **D** Soft cooling gripper composed of LMS-ACE and flexible thermoelectric device (FTED). The FTED is used as the inextensible layer of the gripper, which could be bent and release its hot-side heat when the water is pressurized to flow through the channels of the LMS-ACE. **E** Hydraulically actuated bending of soft gripper and cooling performance. Optical and thermal infrared images show that the circular aluminum block (diameter of 50 mm) with 70 °C is gripped to cool down.

## Thermal-mechanical characteristics of LMS-elastomer (LMSE)

Our LMSE's high performance in both stretching and thermal conduction benefits from its bicontinuous phase (composed of LM network and silicone matrix), which could enable ACE's significant heat charging/discharging effects, especially for large stretching. The topological micro-morphology of LMS is controlled by the infilling ABS pattern (corresponding to the silicone matrix) through FDM printing.

There are five patterns considered here to test their impacts on the LMSE thermal conduction, including the types of Gyroid, Grid, Honeycomb, Cubic, and Concentric (Fig. 2A and Supplementary Fig. 8A). These micro patterns are carried out through layer-by-layer printing with designed infilling paths (Supplementary Movie 3). For the type of Honeycomb (or Cubic), the 3D continuous network of the LMS could be established by filling staggered hexagonal (or square) shapes for

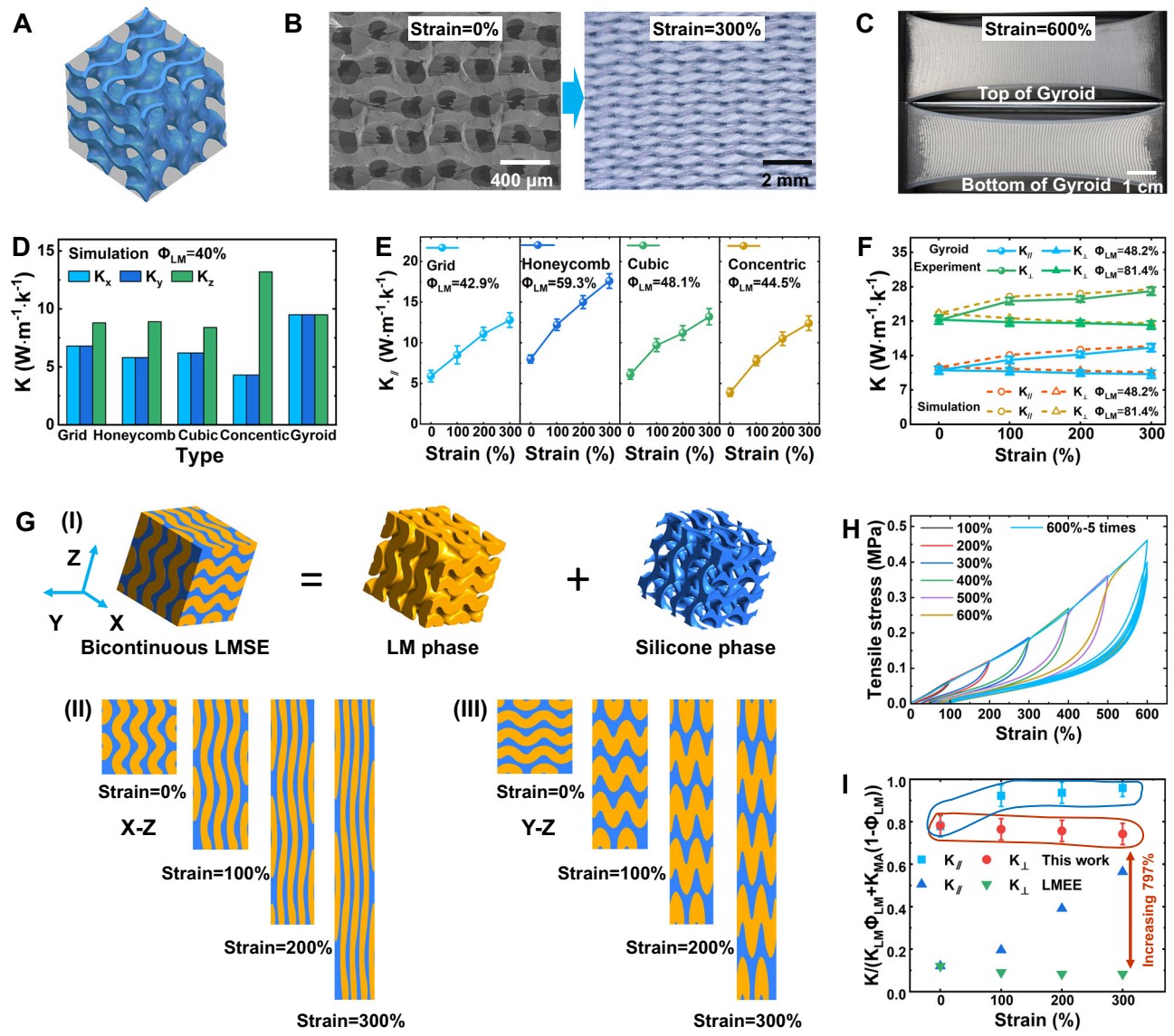

**Fig. 2 | Thermal-mechanical characteristics of liquid metal skeleton-based elastomer (LMSE). A** Illustrations of butadiene styrene (ABS) infilling Gyroid pattern of fused deposition modeling (FDM) printing. **B** Scanning electron microscopic (SEM, strain = 0%, left) and micro-optical (strain = 300%, right) images of Gyroid-type LMS. **C** Optical images of LMSE with Gyroid pattern under strain = 600%. **D** Simulation results of direction-dependent thermal conductivities of $\Phi_{LM}$ = 40% LMSE with different microstructures. **E** Experiment results of thermal conductivities of LMSE with different LMS patterns under stretching, $K_{\parallel}$ for the tensile direction. **F** Experiment and simulation results of thermal conductivities of LMSE with Gyiod type under stretching, $K_{\perp}$ for the perpendicular direction. **G** Simulation results of the Gyroid-type LMSE versus the strain: (I)The simulation geometry model of the LMSE with the bicontinuous structure; (II) The wave-shaped LM line could be straightened when its orientation is consistent with the stretching direction of Z axis; (III) For orientation perpendicularity, the stretching cannot enlarge the gap (filled with silicone elastomer) largely between the LM lines but intensify wave-shape amplitude. **H** Cyclic loading curves of strain-stress for $\Phi_{LM}$ = 81.4% LMSE under different strains, where five cycles are conducted for the strain of 600%. **I** Comparison of thermal conductivity of renormalization for the LMSE and liquid metal-embedded elastomer (LMEE), the LMEE data from ref. 24. Values in Fig. 2E, F, I represent the mean with error bars (standard deviation) (n = 3; independent samples).

adjacent layers, which also weakens its anisotropy. However, the type of Concentric cannot obtain the 3D continuous conduction path of the LMS, which would lead to remarkable anisotropy. The topological microstructures of LMS embedded in the silicone elastomer cannot change by large stretching (strain 300% in Fig. 2B, Supplementary Fig. 8B, and Supplementary Movie 4), constituting the primary thermally conductive pathways even for the strain of 600% (Fig. 2C and Supplementary Fig. 8C). The simulation results have indicated that anisotropy of the micro-morphology (such as Grid, Honeycomb, Cubic, and Concentric) would induce the change of thermal conductivity along different directions (Fig. 2D). For the type of Concentric, its anisotropy leads to $K_z = 12.9$ W/mK and $K_x = K_y = 4.3$ W/mK

for $\Phi_{LM} = 40\%$ under the free strain. The thermal conduction in the z-axis direction follows the parallel rule of $K_z = \Phi_{LM}K_{LM} + (1-\Phi_{LM})K_{MA}$ ($K_z = 13.5$ W/mK, $K_{LM} = 33.4$ W/mK for gallium and $K_{MA} = 0.2$ W/mK for silicone matrix) due to its quasi-2D microstructure. The Gyroid pattern presents the isotropic structure (Supplementary Movie 5) and leads to the homogenous distribution of heat conduction paths ($K_x = K_y = K_z = 9.5$ W/mK for $\Phi_{LM} = 40\%$). The experiment results (Fig. 2E) using the transient hot-wire method[24] (Methods and Supplementary Fig. 9) show that the stretching could enhance the LMSE heat conduction when the LM is elongated and reoriented along with the silicone matrix[33]. For example, the thermal conductivity of the Grid pattern has an increase of 117% along the stretching direction from

zero strain ($K = 5.9$ W/mK) to 300% ($K_{\parallel} = 12.8$ W/mK). The strong anisotropy of the Concentric pattern would induce spatial non-uniformity of thermal conductivity under a large tensile strain. The stretching-enhanced thermal conduction could be observed where the stretching is along the direction of the LM-slab array. Otherwise, the tensile stress would lead the LM-slab array to separate and worsen thermal conduction.

The Gyroid pattern type could achieve a higher thermal conductivity for all the orientations compared with other micro-morphologies of LMS. The large surface area of the Gyroid pattern is beneficial in enhancing the interface interactions with the silicone matrix, which could inhibit the separation of the bicontinuous phases induced by the high LM density. The Gyroid pattern-based LMSE is thus chosen for the subsequent tests and applications. Surprisingly, the Gyroid pattern type achieves a high thermal conductivity of $K_{\parallel} = 27.1$ W/mK in the stretching direction under the strain of 300% for $\Phi_{LM} = 81.4\%$ (Fig. 2F), which approaches the limit of the parallel rule of $K = \Phi_{LM}K_{LM} + (1-\Phi_{LM})K_{MA} = 27.2$ W/mK. Compared with $K = 21.2$ W/mK under the free strain, however, a slight decrease of $K_{\perp} = 20.2$ W/mK in the perpendicular direction is observed. For the type of Gyroid pattern, layer-by-layer staggered wave-shaped LM lines (determined by the ABS infilling patterns, Supplementary Fig. 10) present direction-dependent deformation under the large tensile stress, which would lead to thermal conduction anisotropy but maintain high values. The wave-shaped LM line could be straightened to enhance $K_{\parallel}$ and have less impact on $K_{\perp}$ when its orientation is consistent with the stretching direction (Fig. 2G). The simplified theoretical model also indicates that the stretching could decrease LM skeleton bending (corresponding to wave amplitude) to shorten the thermal path and increase the effective thermal conductivity (Supplementary Fig. 11). For orientation inconsistency (even perpendicularity), the stretching cannot enlarge the gap (filled with silicone elastomer) between the LM lines but intensify wave-shape amplitude, which would weaken $K_{\perp}$ but have less impact on $K_{\parallel}$. Thus, these combined deformation effects of the LMS remarkably enhance heat conduction in the stretching direction, consistent with the numerical results. LMS's deformation features help maintain the silicone matrix's tensile capacity and enable the LMSE with a low elastic modulus (less than 100 kPa, similar to the soft biological tissue) and large stretchability (strain limit >600%), even for the high LM volume fraction of $\Phi_{LM} = 81.4\%$ (Fig. 2H and Supplementary Fig. 12). The thermal conductivities of $K_{\parallel}$ and $K_{\perp}$ of the Gyroid pattern-based LMSE are higher than that of the LMEE with the same $\Phi_{LM}$ under free or large tensile strain (Fig. 2I). More importantly, a high $\Phi_{LM} = 80\%$ of the LMEE would lead to a low strain limit of 75%[30], remarkably lower than 600% of the LMSE. The 3D continuous LMS network also enables the LMSE high volumetric electrical conductivity of $\sigma = 0.64 \times 10^6$ S/cm under free strain and $\sigma = 2.07 \times 10^6$ S/m when strained to 300% for $\Phi_{LM} = 43\%$, respectively (Supplementary Fig. 13).

## Hydraulic-driven adaptive morphing of LMS-ACE

The combination of metal-like thermal conductivity (up to 27.1 W/mK), low stiffness (<100 kPa), and high strain limit (>600%) of the LMSE enables its cooling effects and adaptive morphing capacity. Inspired by the natural active vasodilation principle for blood flow regulation, we present a strategy of hydraulic-driven adaptive morphing to enhance the cooling performance of the LMS-ACE. The soft biological tissue embedded with the vascular networks allows them to control heat removal by dynamically changing shape through constriction or dilation of peripheral vessels. As illustrated in Fig. 3A, the heat from the target cooled object surface through the conductive paths of interactive interface and LMS-ACE matrix arrives into the thermo-fluidic channel, which is then removed by fluid to the radiator and released to the air. Our hydraulic-driven strategy is efficiently conducted by controlling thermo-fluidic pressure through an external pressure controller, which leads to fluidic-channel expansion and enhances the

cooling performance. The sleeve-shaped LMS-ACE with an inner diameter of $D = 21$ mm (Fig. 3B) is used to demonstrate the principle of hydraulic-driven strategy. Both experiment and simulation results show that the sleeve-shaped LMS-ACE could tightly conform to the cylindrical surface with a large diameter (up to 90 mm, Fig. 3C) and even cone body (Supplementary Fig. 14) due to its passive deformation capacity. A large circumferential tensile strain of about 328% of the inner wall of LMS-ACE ($\Phi_{LM} = 81.4\%$) (for $D = 90$ mm) could increase its thermal conductivity to about 27.1 W/mK in the circumferential direction. Although its radial-direction thermal conductivity of about $K_{RD} = 20$ W/mK has minor changes for $D = 90$ mm, the inner wall thickness undergoes a significant reduction from $t = 2$ mm to 0.5 mm (Fig. 3D), leading to a lower thermal conduction resistance of $R_{IW} = t/K_{RD}$ (0.25 cm$^2$K/W). When the inner wall of the LMS-ACE is completely constrained on the body surface, actively increasing the flow channel pressure could create a large tensile strain at its outer wall but almost no influence on the inner-wall deformation (Fig. 3A). The cross-section area of the flow channel increases 214% when $P = 0$ kPa increases 12.5 kPa (Supplementary Fig. 15), allowing more fluid into the channel and decrease the thermal heat-capacity resistance of the flow. This behavior is very similar to vascular vasodilation for blood flow regulation.

Both passive and active conformal deformation of the sleeve-shaped LMS-ACE for the smooth cylindrical surface could induce high contact pressure on the solid-soft interface (Fig. 3E), helping to reduce thermal contact resistance. However, passive deformation cannot achieve seamless contact for the cylindrical surface with large roughness (Fig. 3F), where the residual air gap severely blocks the interface heat transfer. Our active hydraulic-driven strategy successfully covers this critical issue. When the fluid pressure increases to 15 kPa, seamless contact of the solid-soft interface is observed. Notably, the over-loading pressure would lead to the solid-soft interface separation near the channel fins, such as the local contact pressure approaching zero when fluid pressure increases to 20 kPa (Fig. 3F). The main reason is that the fluid pressure would induce the outer wall's outward expansion, create a large tensile strain in the channel fins along the radial direction, and reduce the contact pressure even to cause interface separation. Thus, the fluid pressure of 17.5 kPa is optimal for this case. The zero-contact pressure could also occur on the smooth surface. Six corners of the cross-section with a hexagon shape withstand the most contact pressure, leading to zero contact pressure near each edge's center (Fig. 3G). Increasing the fluid pressure could remarkably enhance the contact pressure near the edge's center but reduce that for the corner, improving the uniformity of the contact stress. For more complex shapes (such as the concave surface), the local fluid pressure control is more suitable for enhancing conformal effects. We only increase the pressure of the single channel near the concave parts rather than all the channels to achieve seamless solid-soft contact (Fig. 3H and Supplementary Movie 6). Both experiment and simulation results demonstrate that the loading fluid pressure larger than 22 kPa meets the requirement (Fig. 3I), which cannot induce the solid-soft interface separation due to the low-pressure loading for other channels.

## Thermal harvesting through LMS-ACE integrated with FTED

The thermoelectric generator-based waste heat recovery is a promising approach to reduce fuel consumption and lower vehicle $CO_2$ emissions, especially for agricultural applications. However, the solid structures of the traditional thermoelectric generator and heat sink are hard to conform to the circular shape of the exhaust pipe[36]. Here, our FTED can adapt to the exhaust pipe (Fig. 4A), and the LMS-ACE is used as the hydraulic-enhanced heatsink on its cold side (Fig. 4B). It is accessible to install them on the tractor for waste heat harvesting during rotary tillage operation (Fig. 4C). The combination of FTED with the LMS-ACE presents desired conformation ability. The simulation

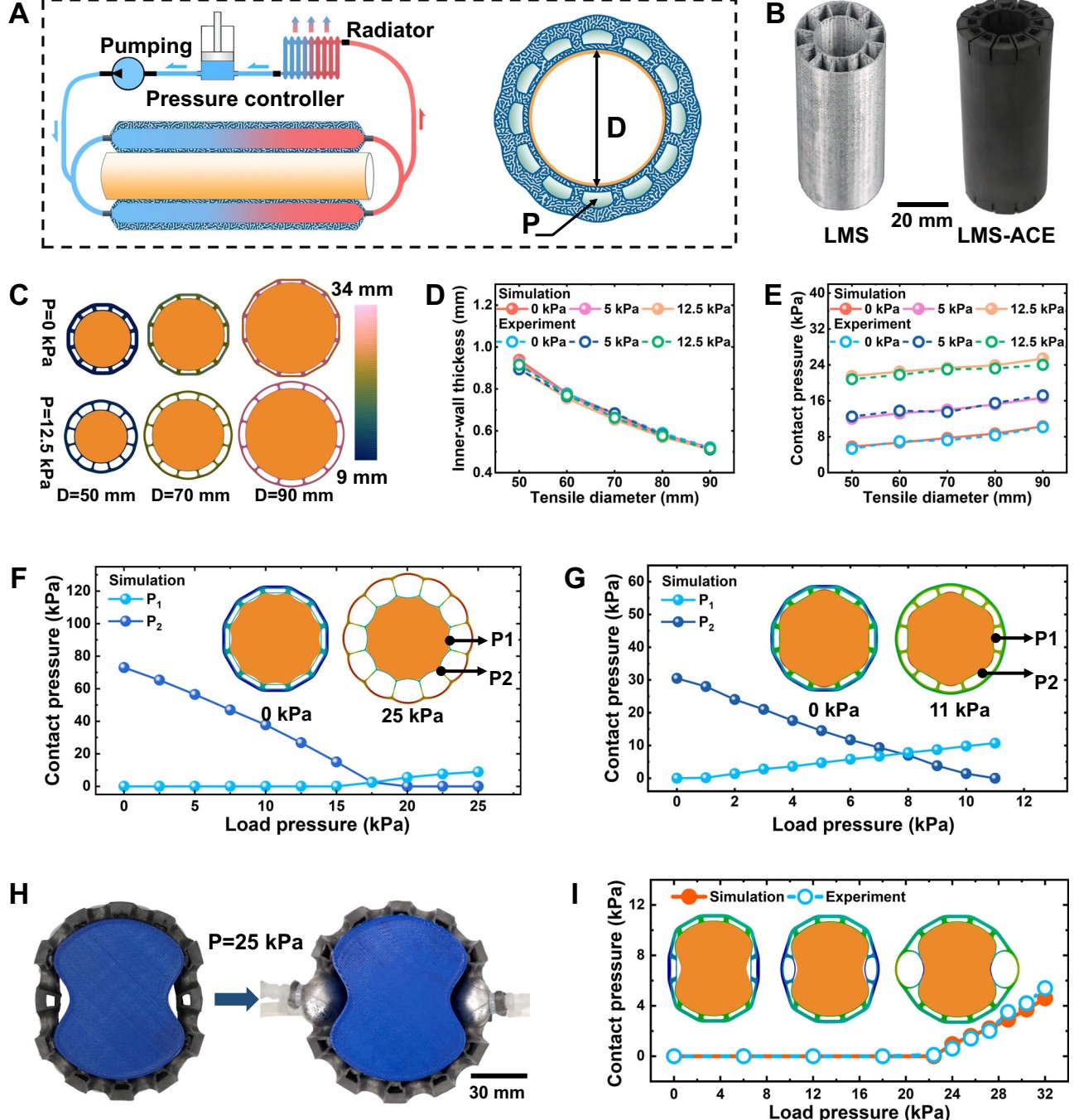

**Fig. 3 | Hydraulic-driven adaptive morphing of liquid metal skeleton-based active-cooling elastomer (LMS-ACE).** **A** Schematic illustration of hydraulic-driven adaptive morphing of LMS-ACE (left) and its cross-section (right). The thermo-fluid is driven by pumping to pass through LMS-ACE (absorbing the heat from the thermal source) and the radiator (releasing the heat to the air environment). The hydraulic pressure of the channel is modulated to increase the fluid flow and enhance the cooling performance of LMS-ACE. **B** Optical images of sleeve-shaped LMS and LMS-ACE (with $D = 21$ mm). **C** The deformation displacement of sleeve-shaped LMS-ACE when it fixes on the cylinder with diameters from 50 mm to 90 mm. **D** The curves of the sleeve-shaped LMS-ACE inner-wall thickness versus tensile diameter. **E** The curves of the sleeve-shaped LMS-ACE inner-wall contact pressure versus tensile diameter. **F** The contact pressures at P1 and P2 would change versus the loading pressure for the cylinder with large roughness. **G** The contact pressures at P1 and P2 would change versus the loading pressure for the cylinder with the cross-section of a hexagon shape. **H** Optical images of LMS-ACE on the concave body with and without loading fluid pressure. **I** Impacts of the loading pressure on the contact pressure of the concave parts.

experiment (Supplementary Fig. 16) results have indicated that the higher thermal conductivity of the LMS-ACE leads to a larger temperature difference ($\Delta T$) between the cold/hot sides of FTED than that for LMEE-ACE (Fig. 4D), which thus creates more output power (Fig. 4E). When the hot-side temperature of FTED is 230 °C, $\Delta T = 163$ °C and *power* = 10.6 W (corresponding to the output power density of

96.0 mW/cm²) are obtained for LMS-ACE, but $\Delta T = 76$ °C and *power* = 2.3 W (20.8 mW/cm²) for LMEE-ACE, which approximately follows the quadratic rule of *power* ~ $(\Delta T)^2$[37]. The output power of FTED could be further improved through the hydraulic-driven expansion of the channel cross-section area (Fig. 4B), which achieves an increase of 13.9% for the loading pressure of 13 kPa (Fig. 4F). We also test the

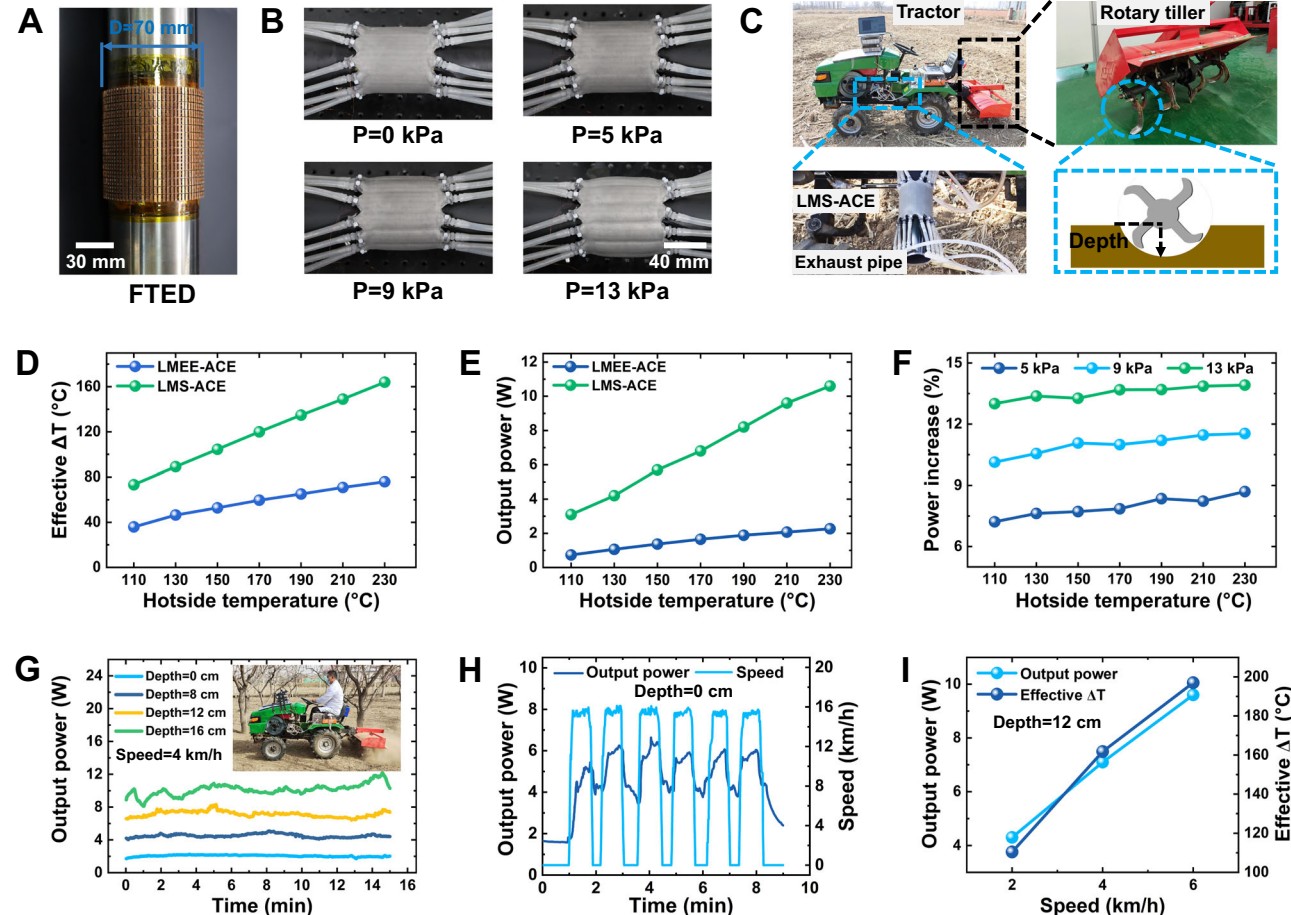

**Fig. 4 | Thermal harvesting through liquid metal skeleton-based active-cooling elastomer (LMS-ACE) integrated with flexible thermoelectric device (FTED).** **A** Optical images of FTED adapted to the exhaust pipe. **B** Optical images of sleeve-shaped LMS-ACE adapted to the FTED on exhaust pipe under different loading hydraulic pressure. **C** Optical images of the tractor and FTED integrated with the sleeve-shaped LMS-ACE for waste heat harvesting during rotary tillage operation. **D** The effective temperature difference between the cold/hot sides of FTED under different hotside temperatures (corresponding to the exhaust air temperature) for LMS-ACE and liquid metal-embedded elastomer (LMEE)-ACE. **E** The curves of output power versus the hotside temperature for LMS-ACE and LMEE-ACE. **F** The output power improvement by the hydraulic-driven strategy with different loading hydraulic pressures. **G** The average output power for different plowing depths under the driving speed of 4 km/h. **H** The impacts of the driving speed on the output power of FTED under the plowing depth of zero. **I** The average output power for different driving speeds under a plowing depth of 12 cm.

performance of the FTED combined with LMS-ACE under the actual operating conditions of the tractor (Supplementary Movie 7). Adding the depth of the rotary tiller (Fig. 4C) would increase the workload of the tractor's fuel engine, thus leading to a higher exhaust temperature. The average output power of 10.4 W (94.2 mW/cm²) is achieved for a plowing depth of 16 cm (Fig. 4G). The workload of the fuel engine is also determined by the driving speed (Supplementary Fig. 17). It could produce a large output power by increasing driving speed even for a depth of zero (Fig. 4H). For a plowing depth of 12 cm, the driving speed of 6 km/h could achieve an average output power of 9.6 W (Fig. 4I). It is noteworthy that increasing the FTED number could improve the output power, which can supply enough energy for many low-power sensors.

### Wearable cooling headband for personalized thermoregulation

Active thermoregulation plays a critical role in human comfort and health, especially in helping keep vital organs of the head nice and toasty. The efficient head-cooling not only protects the brain against the heat stress from the extreme ambient temperature but also reduces the neurological injuries induced by high fever[18] (or even prevents chemotherapy-induced hair loss[38]). However, the existing head-cooling device is bulky and has poor wearing comfort. Based on our

LMS-ACE integrated with the FTED, we design and fabricate an intelligent headband cooling device (Fig. 5A and Supplementary Movie 8) that presents desired flexibility, compactness, lightweight (<65 g, no considering battery), high cooling capacity (temperature drop >20 °C), significant temperature-control accuracy (±0.25 °C), and long-term operation (>24 h). The smart cooling headband mainly consists of a FTED (Fig. 5B), two LMS-ACEs, a water pump, a battery, and a cooling-control circuit, which are integrated into the elastic band. Under the action of direct current, the heat absorption from the forehead through the direct-contact cold side of the FTED is removed to its hot side and arrives into the LMS-ACE (denoted by index I, Fig. 5C). The water driven by the micro-pump circularly flows through the channels in two LMS-ACEs (connected by the pipe) in sequence, which enables the heat release into the ambient air through the surface of LMS-ACEs due to its high thermal conduction. Notably, the LMS-ACE (denoted by index II) bottom adopts pure silicone as the thermal insulation layer to prevent heat from returning to the head (Supplementary Fig. 18). The closed-loop cooling-control circuit is designed to accurately control the forehead temperature (Fig. 5D and Supplementary Fig. 19). The Bluetooth module sends the forehead temperature monitored by the temperature sensor to the mobile phone and receives the preset temperature information from the mobile phone. To achieve the

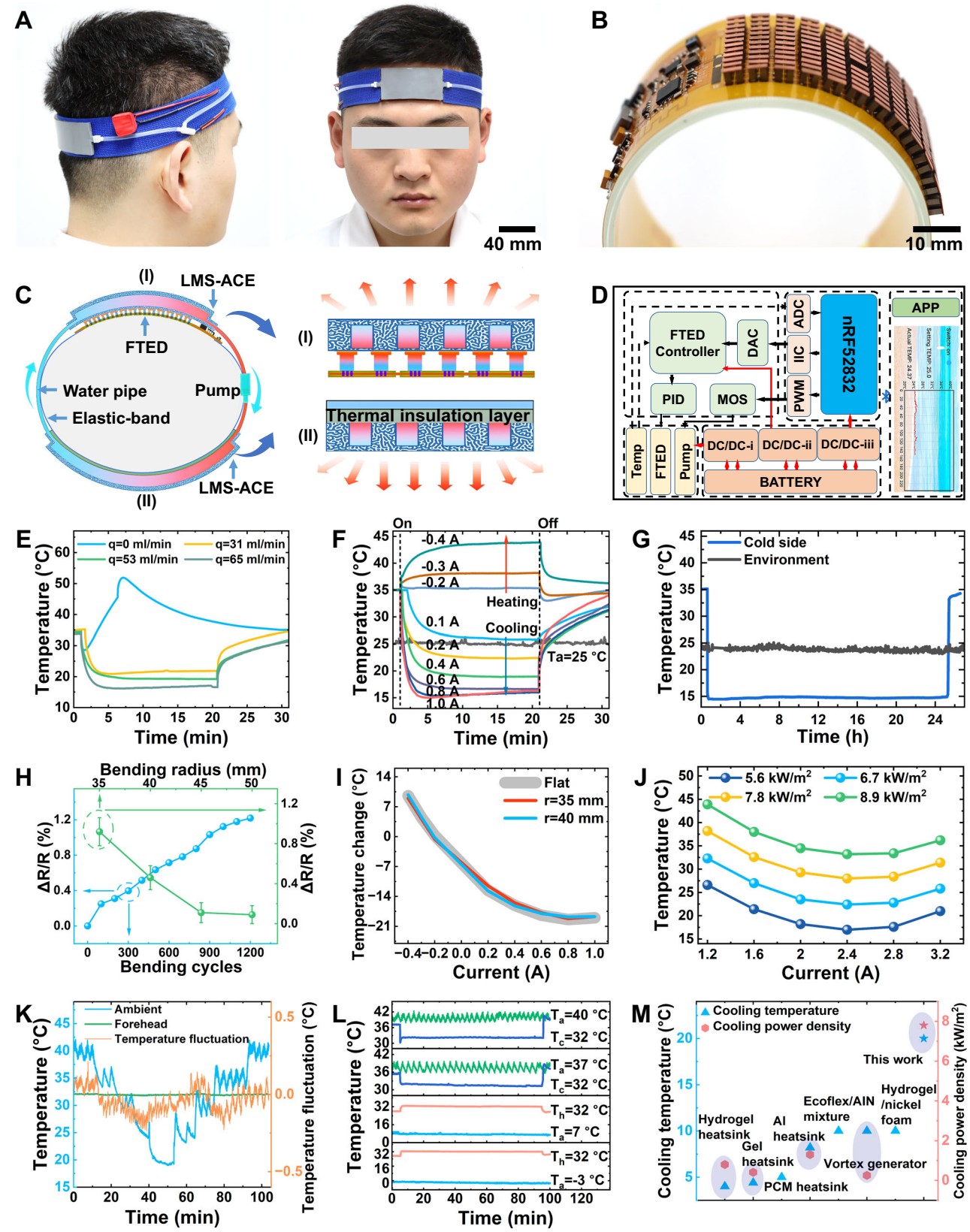

preset temperature, we use the microcontroller unit integrating with a proportional-integral-derivative (PID) strategy to dynamically adjust the cooling performance by changing the input voltage of the FTED. It is thus easy to monitor forehead temperature and adjust it in real time through the mobile phone Application (APP). The LMS-ACE could

effectively remove the heat from the FTED, enabling its desired cooling performance. The surface temperature of the simulated heat source with a heat flux of 324 W/m² (corresponding to the typical metabolic rate from a human under strenuous exercise) has a large drop of 20 °C from the steady-state temperature of 35 °C to about 15 °C (lower than

**Fig. 5 | Wearable cooling headband for personalized thermoregulation.**
**A** Optical images of the cooling headband worn on the head. **B** Optical image of the flexible thermoelectric device (FTED) composed of many P/N-type thermoelectric legs and the control circuit, which are integrated on the flexible film. **C** Schematic illustration of cooling headband working principle. The water is driven by pumping to pass through the first liquid metal skeleton-based active-cooling elastomer (LMS-ACE) (denoted by index I, absorbing the heat from the hotside of FTED) and the second LMS-ACE (denoted by index II, releasing the heat to the air environment). **D** Schematic illustration of cooling-control circuit working principle. **E** Impacts of the LMS-ACE with different water flow rates on the FTED cooling performance. **F** Cooling and heating effects of the FTED modulated by varying input current

direction. **G** Long-term operation test of FTED for controlling temperature (corresponding to the coldside temperature of FTED) of the simulated heat source (324 W/m$^2$). **H** Impacts of FTED bending on its electric resistance. **I** FTED performance comparison under different bending radii. **J** The coldside temperature of FTED versus the input current for different cooling flux densities. **K** Temperature control performance of cooling headband on human forehead under severe change of ambient temperature. **L** Thermoregulation of the human forehead under different ambient temperatures from −3 °C to 40 °C. **M** The cooling performance comparison of this work with the reported work. Values in Fig. 5H represent the mean with error bars (standard deviation) ($n = 3$; independent samples).

the ambient temperature $T_a = 25$ °C) when the flow rate of water is 65 ml/min (Fig. 5E). However, the FTED cooling would fail without heat remove by the LMS-ACE. We can change the current direction to make the FTED cooling or heating and adjust its value to achieve the temperature as needed (Fig. 5F). The FTED combined with LMS-ACE could maintain continuous cooling performance for long-term under the conditions of stable power supply (>24 h, Fig. 5G). The considerable flexibility of the FTED (Fig. 5H) enables its cooling effects unchanged even for a small bending radius of 35 mm (Fig. 5I). If increasing the flow rate of water in the LMS-ACE to 458 ml/min and fixing the inlet temperature of LMS-ACE at 25 °C, we could achieve a high cooling flux density of 8.9 kW/m$^2$ with a hot-side temperature of 33.2 °C (Fig. 5J). The results demonstrate the strong dependence of the FTED cooling performance on the LMS-ACE.

We test the thermoregulation performance of the LMS-ACE on human foreheads without active cooling of the FTED (Supplementary Fig. 20). The results have indicated that LMS-ACE exhibits good heat dissipation capacity, which could achieve a cooling temperature of about 25 °C for the thermal power density of 573.3 W/m$^2$ under the ambient temperature of 21.5 °C. However, it cannot reach a lower temperature than the ambient temperature without active cooling of FTED. We thus test the thermoregulation performance of our cooling headband (integrating with the FTED) on human foreheads under realistic conditions with different ambient temperatures ($T_a$). Although T$_a$ drastically changes between 20 °C and 42 °C, the forehead with the preset temperature of 32 °C always keeps at 32.0 ± 0.25 °C with a low fluctuation (Fig. 4K). The cooling of FTED could keep the forehead temperature to 32.0 at a high $T_a = 40$ °C, and its heating effects could also maintain this preset point at low $T_a = -3$ °C (Fig. 4L). Our cooling headband presents a better cooling performance compared with the reported FTED[20,39–44] (Fig. 4M).

## Discussion

In summary, we have reported a LMS-ACE with high thermal conductivity (up to 27.1 W/mK) and stretchability (strain limit >600% and stiffness <100 kPa) inspired by the bicontinuous structure of bone tissue and designed a strategy of hydraulic-driven adaptive morphing to enhance its cooling effects in light of the natural active vasodilation principle. We have also shown a simple yet versatile method of fabricating complex-shaped LMS-ACE (especially for the geometry with inner completely isolated hollow cavities) using the 3D-printing dissolvable mold and clarified the impacts of the topological micromorphology on its heat conduction. The LMS-ACE could be morphed by modulating the hydraulic pressure of the embedded fluidic channels to conform to the complicated shape, such as the surface with large roughness and even a concave body. Based on LMS-ACE combined with the FTED, we also demonstrated a series of applications, including a soft gripper with hydraulic-driven conjoint functions of actuation and active cooling, thermal-energy harvesting from the curved surface, and wearable smart cooling headband with excellent flexibility and high cooling capacity. Our LMS-ACE can open perspectives for broad applications in clinical practice, personalized thermal management, flexible electronics, and soft robotics.

## Methods
### Materials
The LMs of pure gallium (Ga, melting point at 29.8 °C) for soft gripper and cooking headband and EGa$_{75}$In$_{25}$ (melting point at 15.7 °C) for other applications are considered in this work, respectively. Ga (99.99%) and In (99.99%) were supplied by Zhuzhou Yilong Hung Industrial Co., Ltd. EGa$_{75}$In$_{25}$ was made by mixing 75 wt% Ga with 25 wt% at 200 °C in a vacuum drying oven for one hour. Silicone of Ecoflex 00-30 was obtained from Smooth-On. Dichloromethane (DCM, 99.5%) was supplied by Peking Reagent. Nickel powder (5 μm in diameter, JN-Ni05P) was purchased from Nangong Jinnuo Welding Material Co., Ltd. ABS 3D printing supplies were purchased from Raise 3D ($D = 1.75$ mm).

### Method for LMS-ACE fabrication
The LMS-ACE fabrication process has been presented in Supplementary Movie 1, which includes three steps. (1) The porous ABS-based geometry model designed was printed by the FDM 3D printer (Raise 3D Pro2 plus, Raise 3D Technologies, Inc.) equipped with 0.2 mm brass nozzle. The porous part could be printed with different infill densities from 5% to 100%, the extruded fiber line width of 200 μm, layer height of 150 μm, and printing speed of 60 mm/s. The exposed surface of the porous part is enclosed by a thin wall with two layers. The support part was printed with infill densities of 100%, an extruded fiber line width of 200 μm, a layer height of 20 μm, an extrusion temperature of 290 °C, and a printing speed = 40 mm/s to enhance the interlayer bonding strength, leading to a dense structure. (2) Then, the FDM-printed ABS-based geometry model was immersed in the LM, and the open-porous structure was filled entirely with the LM and quite the contrary for the support part via vacuum for 5 min. Subsequently, the filled model was placed into a refrigerator at −20 °C for 30 min, making the LM freeze. (3) The ABS model was subjected to thorough dissolution in dichloromethane for 24 h, resulting in the complete removal of the ABS and leaving only the LMS structure. (4) The LMS structure was dipped in the liquid silicone for 10 min and then flowed away excess silicone to prepare the LMSE with the bicontinuous structure with coating silicone due to the capillary effect of porous structures. Liquid silicone was prepared by mixing Ecoflex00-30 A and B using an electric mortar with 500 rpm for 5 min. The liquid silicone is diluted by 10% of Silicone Thinner (smooth on) to reduce the viscosity further. The coating thickness can be controlled by adjusting the number of cycle-dipping times, and the coating thickness of each dip of LMSE is approximately 50–100 μm.

### Method for FTED fabrication
According to our previous work[45], the flexible printed circuit board (FPCB) with porous sandwich electrodes is utilized as the flexible substrate for integrating the P/N thermoelectric legs. Firstly, the FPCB is tightly adhered to the polyimide (PI) tape in order to prevent solder paste leakage through the through-hole during welding. Then, a layer of solder paste (Sn$_{42}$Ag$_1$Bi$_{57}$) with a thickness of approximately 0.1 mm is applied onto the FPCB using a stainless-steel template with a grid size of 1.3 mm × 1.3 mm. The P/N-type thermoelectric legs measuring 1.3 mm × 1.3 mm × 1.55 mm are positioned on top of the solder paste

layer using semi-automated surface-mounted technology (SMT, ZB3245TS by Zhejiang Huaqi Zhengbang Automation Technology Co., Ltd). Then, they are soldered in a reflow oven. The Cu-strip electrodes measuring 3.9 mm × 1.3 mm × 25 μm are placed onto the PI tape using SMT system and subsequently coated with a low-temperature solder paste layer. Finally, these Cu-strip electrodes contact the top surface of P/N-type thermoelectric legs and undergo soldering in the reflow oven for approximately fifteen minute. FTED used in the soft cooling gripper has 102 pairs of the P/N thermoelectric leg with a size of 1.3 mm × 1.3 mm × 1.55 mm and a total area of 15.4 mm × 88.3 mm. The FTED used in vehicle waste heat harvesting has 168 pairs of the P/N thermoelectric leg with a size of 1.3 mm × 1.3 mm × 3 mm, and a total area of 27.8 mm × 66.2 mm. The FTED used in wearable Cooling headband has 143 pairs of the P/N thermoelectric leg with a size of 1.3 mm × 1.3 mm × 3 mm, and a total area of 31 mm × 62.3 mm.

## Method for soft gripper fabrication

The soft gripper mainly consists of LMS-ACE used as the extensible layer, and an FTED used as the inextensible layer. LMS-ACE consists of two parts: the LMSE fin (110 mm × 25 mm × 1 mm) and the chambers of the extensible layer. The chambers (110 mm × 25 mm × 10 mm) were prepared by the elastomer compound of silicone matrix and nickel powder by casting them into gripper molds.

## Method for cooling headband fabrication

As shown in Supplementary Movie 8, Supplementary Fig. 18 and Fig. 19, the smart cooling headband mainly consists of an FTED with a temperature sensor (TCTR0402F10K0F3950T) in the middle, two LMS-ACEs, a water pump, a battery, and a cooling-control circuit, which are integrated into the elastic band. The cooling-control circuit has four main modules in the control circuit, including the FTED controller module (ADN8834), Bluetooth module (nRF52832 and DAC8411), DC/DC Convert module (Tps5430, Tps78233, and SE5218ALG-LF), and PUMP module. The LMS-ACEs and pump are connected by silicone tubes, forming a closed loop. Water is injected into the loop through a needle before the test, and no bubbles are generated in the whole loop during the test.

## Microstructure characterization of LMS

The micro-morphology of LMS was obtained by a scanning electron microscopic (SEM, Hitachi SU3500), and the 3D microstructure of LMS with the size of 3 mm was analyzed by the Xradia 410 Versa (Carl Zeiss, resolution ratio = 2.97 μm) micro-computed tomography (Micro-CT), a digital metalloscope (SZ6T) was also utilized to observe the microscopy images of LMS.

## Thermal conductivity measure

The thermal conductivity of LMSE with the sample size of 40 mm × 40 mm × 5 mm was measured by the transient hot-wire method (THW). A platinum wire ($D$ = 76 μm, $L$ = 25 mm) was placed between the two tested samples. The two ends of the wire were soldered to copper and connected to a Keithley 2450 Signal Source Measurement Unit (SMU) in a four-probe configuration. A current pulse (0.9 s) with 500 mA was applied to the Pt wire, and the voltage was measured over 50 data points. The alteration in temperature versus time was derived through computations of resistances subsequently used to approximate the thermal conductivity of LMSE.

## Electrical and mechanical Characteristics

The LMSE conductivity ($\sigma$) is calculated by applying the formula $\sigma = L/RS$, where $L$ represents the length, $R$ denotes the resistance, and $S$ is the cross-sectional area. The LMSE resistance was measured by employing the four-wire method with Keysight 34420 A with the mechanical deformation of the LMSE samples (40 mm × 10 mm × 1 mm) stretched by a linear motor, with the two ends of the LMS electrically connected via copper electrodes. In the tensile test (Instron 5567 Mechanical Testing Machine), the dog-bone-shaped specimens (2 mm × 6 mm × 115 mm) were uniformly stretched at a rate of 20 mm/min.

## Vehicle waste heat harvesting test

The flow of high-temperature engine exhaust gases through the tailpipe was simulated using an exhaust pipe, air blower, and heater, as depicted in Supplementary Fig. 16. Airflow is generated by a blower and heated by a heater connected to the exhaust pipe before being directed into the exhaust pipe itself. The exhaust pipe incorporates an FTED with a sleeve-shaped LMS-ACE (inner diameter D = 21 mm, outer diameter D = 45 mm) consisting of 12 channels as the heatsink for the cold side. To effectively cool the cold side of the FTED, water from a water tank is transferred to the LMS-ACE using a 1.5 W power-rated water pump. Additionally, this pump facilitates the cooling of circulating water through the radiator. A hydraulic-driven strategy with different loading pressures was controlled by an external pressure controller (Y-917). Flow meters were used to measure water flow rates while output power generated by the FTED was tested using an electronic load meter (Itech 8811), and exhaust temperature was measured using Agilent 34,972 A equipment. Furthermore, we evaluated the performance of the FTED combination with LMS-ACE mounted on a tractor's exhaust pipe during its operation in farmland conditions using a tractor model RK150 from Weifang Fusheng Machinery Company. The entire system was powered by the mobile power supply (SXL-02) from Zhongshan Dalji Electric Co., LTD. Tractor speed varied at 2 km/h, 4 km/h, and 6 km/h, corresponding to different tillage depths in farmland: 0 cm, 8 cm, 12 cm, and 16 cm, respectively. We conducted tests under various conditions, including tractor acceleration and deceleration scenarios, tractor climbing situations at different speeds, and flat driving conditions at varying speeds.

## Cooling headband performance test

We built a thermostatic chamber that can set temperatures between 18 °C to 45 °C to simulate the high temperatures. The low-temperature test is carried out in an outdoor environment in cold winter. During the test, the human wore the cooling headband to sit and relax at the test temperature. One thermocouple was placed between the forehead and the FTED to measure the head temperature, and the other thermocouple was placed 10 cm away from the head to measure the actual temperature of the thermostatic chamber. When the head temperature was stable for at least 5 min, the switch was turned on to keep the head temperature at 32 °C under different ambient temperatures. The temperature changes of the head and the thermostatic chamber were recorded by Agilent 34972 A.

## Simulation method

For further details on the simulation methods, see Supplementary Information Note 1.

# Data availability

All data are available in the main text and Supplementary Information. Source data are provided as a source data file. Source data are provided with this paper.

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

## Acknowledgements

The authors would like to acknowledge the National Natural Science Foundation of China (NSFC) (Grant No.52076213) and the 2115 Talent Development Program of China Agricultural University for the financial coverage of this work.

## Author contributions

Z.H. supervised the investigations. D.Y. and Z.W. designed and con-ducted the experiments. G.C. performed the numerical simulation. Q.Z., J.F., M.L. and C.L. conducted the experiments of FTED integrated with LMS-ACE. Q.Z., Z.L., D.C. and Z.S. helped analyze data and provided the experiment platform of the tractor. D.Y. drafted the manuscript with inputs from all other authors. All the authors agreed on the final manuscript.

## Competing interests

The authors declare no competing interests.

## Ethics

All procedures for the cooling/heating headband testing and demonstrations involving human participants are approved by Human and Artefacts Ethics Committee, China Agricultural University, with the reference number CAUHR-20231208. The informed consent of all participants was obtained prior to inclusion in this study.
