## [Peer Review File · Nature Communications]

Hydraulic-driven adaptable morphing active-cooling elastomer with bioinspired bicontinuous phasesREVIEWER COMMENTS

Reviewer #1 (Remarks to the Author):

In this manuscript, the authors present the fabrication of the liquid metal skeleton-embedded elastomer and the application of active cooling. The authors developed the fabrication method of the 3D printing of molding template and subsequent infiltration of liquid metal and coating for producing the highly thermally conductive and mechanical stretchable elastomeric materials, which was further supported by the thermal conductivity measurement under stretching. Moreover, the authors demonstrated the hydraulic-driven morphing for conformal thermal contact and active cooling of thermoelectric devices. My impression after a thorough reading is that the work is publishable in Nature Communications since the subject matter can potentially appeal to a broad audience in the thermal engineering and thermoelectric community. It is interesting that the 3D-printed architecture was used as a sacrificial template, which enabled the formation of complex architectures of liquid metal skeletons. The demonstration of cooling combined with thermoelectric devices further supports the practicability of this work. The manuscript is well organized and the data supports the conclusion. Some issues are presented below.

1. The liquid metal skeleton-embedded elastomer was constructed by the liquid metal skeleton and silicone rubber coating layer. What was the thickness of the silicone layer? This is an important parameter that can affect the thermal transport property due to its ultralow thermal conductivity.
2. The authors claimed the enhancement of thermal conductivities of the samples under stretching and suggested the mechanism in that the LM is elongated and reorientated along with the silicone matrix under stretching. This is not understandable considering that the elongation and reorientation of macroscopic architectures can't affect the intrinsic thermal transport. The authors should note that this is not the case like the enhancement of electrical properties of electrically conductive elastomers under stretching, in which conductive metallic nanostructures like nanowires, and flakes are embedded. In these materials, the conductive nanostructures are elongated and oriented under stretching so that the intrinsic electrical transport properties are enhanced by the structural changes. However, there is no chance of such nanostructural changes for liquid metals. I suspect the changes in the volume fraction or the oxide skin breakage of liquid metals under stretching, which can change the thermal conductivity.
3. The very high electrical conductivity is not understandable because the liquid metal skeletons are coated by the insulating silicone layer. This high property can be attributed to the contacting of the probe to the interior liquid metal directly when the electrical transport was measured. In this case, the measured properties are not the properties of the samples but the liquid metals.
4. I recommend the demonstration of a wearable cooling headband with the use of only LMS-ACE (no thermoelectric cooler) since it can exhibit excellent heat dissipation as itself.

Reviewer #2 (Remarks to the Author):

Dear Authors, Dear Editor

The manuscript presents an interesting concept of a porous elastomer with liquid metal, which combines large strain and high thermal conductivity. The manuscript presents in detail the fabrication of such elements and its application in thermoregulation. The authors further demonstrated application of such element in combination with thermoelectric modulus in active cooling and energy harvesting. However, I would suggest the following corrections before the paper can be considered for publication.

1. The figures are very complex and hard to follow. Some of the are not even explained in the text. I would suggest reducing the number of segments in the figures. However, especially the graphs (diagrams) and the trends shown of them should be explained more in detail. In addition, the text on the graphs should be increased.
2. It is stated in page 1, line 32 that K represents cooling coefficient. However, based on the Newton's law of cooling, K is heat transfer coefficient. The authors should thus rename cooling coefficient with heat transfer coefficient.

3. It is not clear why LMEE has more than 10-times lower thermal conductivity compared to LMS-ACE developed in this work.
4. In page 4, line 140, the simulation results are mentioned but nothing about the methodology of these simulations can be found in the manuscript.
5. Some of the elastomers exhibit also the elastocaloric effect. Did the authors notice any temperature changes of the elastomer upon stretching?

Reviewer #3 (Remarks to the Author):

The article "Hydraulic-driven Adaptable Morphing Active-cooling Elastomer with Bioinspired Bicontinuous Phases" introduces a new approach to creating elastomers with liquid metal skeletons for direct contact cooling. The novelty of the approach is that the elastomers are capable of adjusting their shape to the object being cooled, using hydraulic-driven adaptive morphing techniques inspired by blood flow regulation principles, and they demonstrate high stretchability and high thermal conductivity. The article also provides three examples showcasing the versatility of this elastomer when combined with a flexible thermoelectric device. These examples include a soft gripper, a flexible thermal energy harvesting device, and a cooling headband.

The topic is relevant and worth exploring. The manuscript is well written, with high attention to detail and sound methodology. The scope of the paper is very broad.

However, the wide scope of the paper makes it challenging to follow in certain areas. Also, the authors use a lot of abbreviations across the paper, and in my personal opinion, it is a bit too much. There is no list of these abbreviations, so finding the meaning of the specific one takes time. Some abbreviations, such as FTED and FDM, are not explained.

There are four figures in the main text, each containing 15 to 25 subfigures. While the figures are neat, it would be a good idea to move some of the subfigures to supplementary material to make them less crowded and focused on the most important findings. I recommend moving at least Fig. 1 D-F, Fig. 2 - results for patterns different than gyroid, and Fig. 3 (K) to the supplementary material. Additionally, the word "Experiment" is misspelled in Fig. 2 (F) and Fig. 3 (D, E).

**Hydraulic-driven Adaptable Morphing Active-cooling Elastomer with
Bioinspired Bicontinuous Phases
NCOMMS-23-42668
Reply to Reviewers' Comments**

By Dehai Yu, Zhonghao Wang, Guidong Chi, Qiubo Zhang, Junxian Fu, Maolin Li,
Chuanke Liu, Quan Zhou, Zhen Li, Zhe Xin, Du Chen, Zhenghe Song, Zhizhu He

General Response

We would like to thank the reviewers for their constructive remarks and input that helped us further improve the manuscript. We have conducted additional experiments (including **Supplementary Figs. 5 and 20**) and theoretical model (**Supplementary Fig. 11**) for addressing the comments from the reviewers. We also rearranged the figure layout and added more descriptions to improve the manuscript's readability. Point-to-point responses to each comment are listed in the following, with corresponding changes highlighted in the revised manuscript in **blue** for easy tracking.

Response to Reviewer 1:

Summary Comment: In this manuscript, the authors present the fabrication of the liquid metal skeleton-embedded elastomer and the application of active cooling. The authors developed the fabrication method of the 3D printing of molding template and subsequent infiltration of liquid metal and coating for producing the highly thermally conductive and mechanical stretchable elastomeric materials, which was further supported by the thermal conductivity measurement under stretching. Moreover, the authors demonstrated the hydraulic-driven morphing for conformal thermal contact and active cooling of thermoelectric devices. My impression after a thorough reading is that the work is publishable in Nature Communications since the subject matter can potentially appeal to a broad audience in the thermal engineering and thermoelectric community. It is interesting that the 3D-printed architecture was used as a sacrificial template, which enabled the formation of complex architectures of liquid metal skeletons. The demonstration of cooling combined with thermoelectric devices further supports the practicability of this work. The manuscript is well organized and the data supports the conclusion. Some issues are presented below.

Response: We would like to thank the reviewer for the positive comments. Your constructive suggestions help us significantly improve this work and make it more solid. We have conducted new experiments and theoretical model to address your concerns. In the following, we would like to explain more about your concerns.

Comment 1: The liquid metal skeleton-embedded elastomer was constructed by the liquid metal skeleton and silicone rubber coating layer. What was the thickness of the silicone layer? This is an important parameter that can affect the thermal transport property due to its ultralow thermal conductivity.

Response: Thank you for this comment. We have measured the thickness of the silicone layer under different dipping numbers, as shown in Fig. R1A. The dipping number of 5 for the pure silicone leads to a coating thickness of 0.27 mm. We also used LMEE (with 80 wt% gallium LM, achieving 0.84 W/mK at the zero strain and 3.3 W/mK along the stretching direction at the strain of 300%) to coat the LMS to improve

its thermal conduction. For the actual application, the pre-stretching can remarkably reduce the coating thickness. As shown in **Fig. 1RB**, the LMS (with a thickness of 0.4 mm) is coated with silicone (a coating thickness 0.3 mm) to obtain a thickness of 1 mm, which reduces to 0.3 mm (a coating thickness of about 0.09 mm) at a strain of 300%. Such small thicknesses cannot lead to a large impact on the thermal transport.

Figure R1. Coating layer thickness of LMS. (A) Impacts of the dipping number on the coating layer thickness for the coating materials of pure silicone or LMEE (80 wt% LM). (B) LMS (with a thickness of 0.4 mm) is coated with silicone (a thickness of 0.3mm) to obtain a thickness of 1 mm, which reduces to 0.3 mm (the coating thickness of about 0.09 mm) at a strain of 300 %.

Comment 2: Thermal conductivities of the samples under stretching and suggested the mechanism in that the LM is elongated and reorientated along with the silicone matrix under stretching. This is not understandable considering that the elongation and reorientation of macroscopic architectures can't affect the intrinsic thermal transport. The authors should note that this is not the case like the enhancement of electrical properties of electrically conductive elastomers under stretching, in which conductive metallic nanostructures like nanowires, and flakes are embedded. In these materials, the conductive nanostructures are elongated and oriented under stretching so that the intrinsic electrical transport properties are enhanced by the structural changes. However, there is no chance of such nanostructural changes for liquid metals. I suspect the changes in the volume fraction or the oxide skin breakage of liquid metals under stretching, which can change the thermal conductivity.

Response: Thank you for this comment. Our simulation and experiment results indicate that the stretching-enhanced thermal conductivity in the stretching direction is mainly attributed to the elongation and reorientation of the LM skeleton (**Fig. R2**). For the Gyoid-type LM skeleton (**Fig. R2A**), the simulation results (**Fig. R2B**) indicate that the wave-shaped LM skeleton is straightened along the stretching direction, which is

consistent with the experiment observation (Fig. R2C). The thermal conductivities of LMSE from the simulation and experiment results are also highly consistent (Fig. R2D), which are considerably better than the thermal conductivities of LMEE (Fig. R2E).

Figure R2. Stretching-enhanced thermal conductivity of LMSE. (A) Illustrations of Gyroid-type LMSE constitution. (B) The wave-shaped LM line could be straightened when its orientation is consistent with the stretching direction of Z axis. (C) SEM (strain=0%, up) and micro-optical (strain=300% by stretching in horizontal direction, bottom) images of Gyroid-type LMSE. (D) Experiment and simulation results of thermal conductivities of LMSE with Gyoid type under stretching. K_{\perp} for the perpendicular direction. (E) Comparison of thermal conductivity of renormalization for the LMSE and LMEE, the LMEE data from Ref. (24). (All the sub-figures of Figure R2 are from Figure 2 in the revised manuscript).

In the revised manuscript, we added a simple theoretical model to explain the mechanism of the stretching-enhanced thermal conductivity in the stretching direction for LMSE. As shown in Fig R3 (corresponding to Fig. S11 in the revised manuscript), the wave-shaped segment of the Gyoid-type LM skeleton is straightened along the stretching direction. For the segment length of L and cross-section area of A (including the LM skeleton cross-section area of A_{LM} , indicating the volume fraction of LM with $\Phi_{LM}=A_{LM}/A$), the relation between the thermal flux (Q) and the temperature difference (ΔT) can be used to estimate the effective thermal conductivity (K) of LMSE. To simplify the theoretical analysis, we assume that the thermal conduction of LMSE with a large content of the LM ($\Phi_{LM}>40\%$) is mainly determined by the LM skeleton due to the low heat conductivity of the silicone matrix ($K_{MA}=0.2$ W/mK) compared with LM ($K_{LM}=33.4$ W/mK for gallium). In addition, the volume fraction of LM is kept the same for the two cases (before and after stretching). It is noteworthy that the thermal conduction path length of the wave-shaped LM skeleton is the curve length of S for the segment length of L . The simplified theoretical model indicates that increasing the bending (corresponding to wave amplitude of h) of the LM skeleton would extend the

thermal path and decrease the effective thermal conductivity of K_{wave} . Thus, the wave-shaped segment of the LM skeleton is straightened along the stretching direction, leading to a decrease in h and an increase in K_{wave} . It is noteworthy that the upper limit value of K_{wave} is K_{line} (corresponding to $h=0$). The theoretical model is consistent with the numerical and experiment results. For example, $K_{wave}=27.1$ W/mK (for $\Phi_{LM}=81.4\%$, **Fig. R2C**) approaches to $K_{line}=\Phi_{LM}K_{LM}+(1-\Phi_{LM})K_{MA}=27.2$ W/mK under the strain of 300% ($h\approx 0$, **Fig. R2B**).

Figure R3. A simple theoretical model to explain the mechanism of the stretching-enhanced thermal conductivity in the stretching direction for LMSE.

Comment 3: The very high electrical conductivity is not understandable because the liquid metal skeletons are coated by the insulating silicone layer. This high property can be attributed to the contacting of the probe to the interior liquid metal directly when the electrical transport was measured. In this case, the measured properties are not the properties of the samples but the liquid metals.

Response: Thank you for this comment. As you commented, the high electrical conductivity is attributed to contacting the probe to the interior LM skeleton network directly when the electrical transport was measured. The effective conductivity (σ) is calculated by applying the formula $\sigma = L/RS$, where L represents the length, R denotes the resistance, and S is the cross-sectional area of the LMSE (while not the size of the LM skeleton). We listed the interior electrical conductivity of the LMSE to highlight its potential application as a stretchable material of electromagnetic interference (EMI)

shielding for flexible electronics.

Comment 4: I recommend the demonstration of a wearable cooling headband with the use of only LMS-ACE (no thermoelectric cooler) since it can exhibit excellent heat dissipation as itself.

Response: Thank you for this comment. We have tested the performance of a wearable cooling headband with the use of only LMS-ACE (Figure R4, corresponding to Fig. S20 in the revised manuscript). The results have indicated that LMS-ACE exhibits good heat dissipation capacity, which could achieve a cooling temperature of about 25 °C for the thermal power density of 573.3 W/m² under the ambient temperature of 21.5 °C (Fig. R4C). However, it cannot achieve a lower temperature than the ambient temperature without active cooling of FTED. In addition, the temperature control accuracy of the cooling headband with the use of only LMS-ACE is lower than that when integrating with FTED (Fig. R4E and Fig. 5K in the revised manuscript).

Figure. R4. Performance of the wearable cooling headband with only LMS-ACE without FTED. (A) Photograph of the wearable cooling headband. (B) Photograph of the wearable cooling headband worn on mannequin. The heating film (with the size of 30 mm × 60 mm, the thermal power density of 573.3 W/m²) was attached to the mannequin head. The cooling headband was attached to the heating film, and a thermocouple was placed between the heating film and the cooling headband to obtain real-time temperature data. (C) Long-term operation test of the headband cooling performance. Without the cooling headband, the model's temperature reached 38.1 °C. After wearing the cooling headband, the mannequin's head temperature was reduced from 35 °C to 24.9 °C. (D) Optical images of the cooling headband worn on the head. (E) The cooling performance of headband wearing on the human head. The temperature of the human forehead is 33.0 °C without cooling headband. The cooling headband could maintain the forehead temperature between 28 °C and 29 °C under different conditions of sitting, walking, and running.

Response to Reviewer 2:

Summary Comment: The manuscript presents an interesting concept of a porous elastomer with liquid metal, which combines large strain and high thermal conductivity. The manuscript presents in detail the fabrication of such elements and its application in thermoregulation. The authors further demonstrated application of such element in combination with thermoelectric modulus in active cooling and energy harvesting. However, I would suggest the following corrections before the paper can be considered for publication.

Response: We would like to thank the reviewer for the positive comments. Your constructive suggestions help us significantly improve this work to be more solid and readable. In the following, we would like to explain more about your concerns.

Comment 1: The figures are very complex and hard to follow. Some of the are not even explained in the text. I would suggest reducing the number of segments in the figures. However, especially the graphs (diagrams) and the trends shown of them should be explained more in detail. In addition, the text on the graphs should be increased.

Response: Thank you for this comment. Following your constructive suggestions, we have rearranged the figure layout and added more descriptions to improve the manuscript's readability.

In the revised manuscript, we have removed Fig. 1 D-F to Fig. S6, and Fig. 2 A-C (the results for patterns different than gyroid) to Fig. S8. We have split Fig. 3 into Fig.3 and Fig. 4. In addition, we added more descriptions to figures in the text.

Comment 2: It is stated in page 1, line 32 that K represents cooling coefficient. However, based on the Newton's law of cooling, K is heat transfer coefficient. The authors should thus rename cooling coefficient with heat transfer coefficient.

Response: Thank you for this comment. Following your constructive suggestion, we have renamed "cooling coefficient" with "heat transfer coefficient".in the revised manuscript.

Comment 3: It is not clear why LMEE has more than 10-times lower thermal conductivity compared to LMS-ACE developed in this work.

Response: Thank you for this comment. As shown in **Fig. R1**, the LMEE is obtained by filling the LM micro-droplets into the elastomer, which cannot form the LM thermal path like LMSE (developed in this work). Thus, the thermal conductivity of LMEE is only 1.6 W/mK for a large LM volume fraction of $\Phi_{LM}=50\%$ under stress-free conditions, which even has a slight decrease (<1.6 W/mK) in the orthogonal directions when strained to 400%.

Figure R1. Illustration of the LMEE and LMSE micro-structures.

Comment 4: In page 4, line 140, the simulation results are mentioned but nothing about the methodology of these simulations can be found in the manuscript.

Response: Thank you for this comment. The detailed numerical methodology is given in Supplement Note 1

Comment 5: Some of the elastomers exhibit also the elastocaloric effect. Did the authors notice any temperature changes of the elastomer upon stretching?

Response: Thank you for this comment. In this work, we used the silicone elastomer of Ecoflex 00-30 (Smooth-On), and could not observe the elastocaloric effect during the stretching process (See Fig. R2).

Figure R2. Thermal infrared images of the LMSE under different strains.

Response to Reviewer 3:

Summary Comment: The article “Hydraulic-driven Adaptable Morphing Active-cooling Elastomer with Bioinspired Bicontinuous Phases” introduces a new approach to creating elastomers with liquid metal skeletons for direct contact cooling. The novelty of the approach is that the elastomers are capable of adjusting their shape to the object being cooled, using hydraulic-driven adaptive morphing techniques inspired by blood flow regulation principles, and they demonstrate high stretchability and high thermal conductivity. The article also provides three examples showcasing the versatility of this elastomer when combined with a flexible thermoelectric device. These examples include a soft gripper, a flexible thermal energy harvesting device, and a cooling headband. The topic is relevant and worth exploring. The manuscript is well written, with high attention to detail and sound methodology.

Response: We would like to thank the reviewer for several constructive suggestions, which helped us significantly improve this work. In the following, we would like to explain more about your concerns.

Comment 1: The scope of the paper is very broad. However, the wide scope of the paper makes it challenging to follow in certain areas. Also, the authors use a lot of abbreviations across the paper, and in my personal opinion, it is a bit too much. There is no list of these abbreviations, so finding the meaning of the specific one takes time. Some abbreviations, such as FTED and FDM, are not explained.

Response: Thanks for this comment. We have added a list of these abbreviations in the manuscript for query. We also checked all the abbreviations in the manuscript thoroughly to ensure them be explained.

Abbreviations explanation. ACE: active-cooling elastomer; LM: liquid metal; LMS: LM skeleton; LMSE: LMS-elastomer; LMEE: LM-embedded elastomer; FDM: fused deposition modeling; ABS: acrylonitrile butadiene styrene; FTED: flexible thermoelectric device; DCM: dichloromethane; PID: proportional-integral-derivative; APP: Application.

Comment 2: There are four figures in the main text, each containing 15 to 25

subfigures. While the figures are neat, it would be a good idea to move some of the subfigures to supplementary material to make them less crowded and focused on the most important findings. I recommend moving at least Fig. 1 D-F, Fig. 2 - results for patterns different than gyroid, and Fig. 3 (K) to the supplementary material. Additionally, the word “Experiment” is misspelled in Fig. 2 (F) and Fig. 3 (D, E).

Response: Thanks for this comment. In the revised manuscript, we have removed Fig. 1 D-F to Fig. S6, and Fig. 2 A-C (the results for patterns different than gyroid) to Fig. S8. We have split Fig. 3 into Fig.3 and Fig. 4. We added more descriptions to figures in the text. We also checked the manuscript thoroughly to ensure all the descriptions were correct and readable.

REVIEWERS' COMMENTS

Reviewer #1 (Remarks to the Author):

The authors addressed all of the concerns raised by the reviewer. I recommend the publication.

Reviewer #2 (Remarks to the Author):

Dear Editor; Dear Authors.

All my previous comments are correctly addressed.

Reviewer #3 (Remarks to the Author):

The manuscript entitled "Hydraulic-driven Adaptable Morphing Active-cooling Elastomer with Bioinspired Bicontinuous Phases" has been significantly improved, and all reviewer's comments have been addressed. In my opinion, the paper can be accepted in its current form.

**Hydraulic-driven Adaptable Morphing Active-cooling Elastomer with
Bioinspired Bicontinuous Phases
NCOMMS-23-42668
Reply to Reviewers' Comments**

By Dehai Yu, Zhonghao Wang, Guidong Chi, Qiubo Zhang, Junxian Fu, Maolin Li,
Chuanke Liu, Quan Zhou, Zhen Li, Zhe Xin, Du Chen, Zhenghe Song, Zhizhu He

Response to Reviewer 1:

Summary Comment: The authors addressed all of the concerns raised by the reviewer. I recommend the publication.

Response: We sincerely appreciate the time and effort you dedicated to reviewing our manuscript.

Response to Reviewer 2:

Summary Comment: All my previous comments are correctly addressed.

Response: We sincerely appreciate the time and effort you dedicated to reviewing our manuscript.

Response to Reviewer 3:

Summary Comment: The manuscript entitled “Hydraulic-driven Adaptable Morphing Active-cooling Elastomer with Bioinspired Bicontinuous Phases” has been significantly improved, and all reviewer’s comments have been addressed. In my opinion, the paper can be accepted in its current form.

Response: We sincerely appreciate the time and effort you dedicated to reviewing our manuscript.